

# Environmental factors affecting honey bees (*Apis cerana*) and cabbage white butterflies (*Pieris rapae*) at urban farmlands

Myung-Bok Lee

Institute of Zoology, Guangdong Academy of Sciences, Guangzhou, China

## ABSTRACT

Rapid urbanization results in a significantly increased urban population, but also the loss of agricultural lands, thus raising a concern for food security. Urban agriculture has received increasing attention as a way of improving food access in urban areas and local farmers' livelihoods. Although vegetable-dominant small urban farmlands are relatively common in China, little is known about environmental factors associated with insects that could affect ecosystem services at these urban farmlands, which in turn influences agricultural productivity. Using Asian honey bee (*Apis cerana*) and cabbage white butterfly (*Pieris rapae*) as examples, I investigated how environmental features within and surrounding urban farmlands affected insect pollinator (bee) and pest (butterfly) abundance in a megacity of China during winters. I considered environmental features at three spatial scales: fine (5 m-radius area), local (50 m-radius area), and landscape (500 m-raidus and 1 km-radius areas). While the abundance of *P. rapae* increased with local crop diversity, it was strongly negatively associated with landscape-scale crop and weed covers. *A. cerana* responded positively to flower cover at the fine scale. Their abundance also increased with local-scale weed cover but decreased with increasing landscape-scale weed cover. The abundance of *A. cerana* tended to decrease with increasing patch density of farmlands within a landscape, *i.e.*, farmland fragmentation. These results suggest that cultivating too diverse crops at urban farmlands can increase crop damage; however, the damage may be alleviated at farmlands embedded in a landscape with more crop cover. Retaining a small amount of un-harvested flowering crops and weedy vegetation within a farmland, especially less fragmented farmland can benefit *A. cerana* when natural resources are scarce.

# INTRODUCTION

Urban expansion in many countries is often accompanied by the loss of agricultural lands (*Bren d'Amour et al., 2017*). The world's urban areas are a quarter of total agricultural lands; however, urban expansion has occurred faster than urban population growth and takes place on productive agricultural lands, leading to reduction in global crop production (*Seto & Ramankutty, 2016*; *Bren d'Amour et al., 2017*). Moreover, the proportion of global

Corresponding author
Myung-Bok Lee,
bok.ecology@outlook.com

population in urban areas is forecast to increase from 55% in 2018 to 68% by 2050 (*United Nations, 2019*). Approximately 90% of the increase would occur in Asia and Africa, which are also the hotspot of future urban expansion. These changes, *i.e.,* growing urban population and agricultural land loss, raise concerns for not only food security but also for the livelihoods of smallholders, especially in developing countries (*IFPRI, 2017*; *Huang et al., 2020*).

Urban agriculture has received increasing attention as one of the practices that may improve the accessibility of food, the livelihood of local people such as farmers and rural migrants, and human well-being (*Deelstra & Girardet, 2000*; *De Bon, Parrot & Moustier, 2010*; *Mok et al., 2013*; *Martellozzo et al., 2014*). One recent study also shows that small-scale urban agriculture can bring high yields (*McDougall, Kristiansen & Rader, 2018*). Urban agriculture can be broadly defined as agricultural production, such as vegetables, fruits, ornamental plants, and other dietary products, occurring in urban environments that include both inner city and city fringe areas (*Mougeot, 2000*). In China, small-scale vegetable farming is common in cities, especially at vacant lots and city fringe areas. Urban agriculture, also called peri-urban agriculture in some cases, has been supported by the Chinese government policy "Vegetable Basket" for decades (*Zhong et al., 2021*). Like other countries, urban expansion has been a major driver causing farmland loss in China (*Tu et al., 2021*). Although farmland protection policy started in mid-1990s has alleviated the loss to some extent, urban development frequently occurs on highly productive farmlands near urban edge areas, often leaves reclaimed but less productive farmlands as an offset for the development, and increases farmland fragmentation (*Liang et al., 2015*; *Huang, Du & Castillo, 2019*; *Tu et al., 2021*). In this situation, understanding ecological factors that could affect agricultural production at urban farmlands is crucial to promote the sustainability of these farmlands.

Insects are well known for their role in ecosystem service (*e.g.,* pollination and pest control) and disservice (*e.g.,* crop damage) in agricultural landscapes (*Losey & Vaughan, 2006*; *Kremen & Chaplin-Kramer, 2007*; *Oliveira et al., 2014*; *Omkar, 2018*). Their effects on crop yields have been widely studied. Over 70% of main global crops benefit from animal-mediated pollination (*Klein et al., 2007*) and 9–25% of staple food crops are estimated to be lost due to animal pests (*Oerke, 2006*), in which insects are the major group in both cases. Bees are considered the most important pollinators. For example, in the USA, 11% of the agricultural gross domestic product in 2009 depended on pollination, mainly contributed by bees (*Lautenbach et al., 2012*). Bee abundance affects yields of certain crops such as oilseed rape more than pesticides do (*Catarino et al., 2019* and references therein). In particular, it is widely recognized that honey bees provide important pollination services for a variety of crops although pollination efficiency of the Western honey bees (*Apis mellifera*) in a natural habitat is somewhat debatable (*Hung et al., 2018*). A number of factors, *e.g.,* quantity and quality of floral resources, semi-natural vegetation, crop diversity such as crop compositional or configurational heterogeneity and landscape complexity, influence the abundance and diversity of bees (*Klein et al., 2012*; *Scheper et al., 2013*; *Potts et al., 2003*; *Priyadarshana et al., 2021*; *Raderschall et al., 2021*).

In contrast to bees, cabbage white butterflies (*e.g.*, *Pieris rapae* and *P. brassicae*) are recognized as agricultural pests because their larvae, cabbageworms, feed on the family Brassicaceae crops (cruciferous crops) and can severely damage these crops. For example, cabbage white butterflies caused about 70% yield loss in cruciferous crop in Meghalaya, the northeastern state of India (*Singh, Satyanarayana & Peshin, 2014*). While presence or abundance of Brassicaceae crops is positively associated with cabbageworm density, this effect may be amplified if the host plants are congregated as a large pure stand (resource concentration hypothesis; *Root, 1973*) or sparsely distributed (resource diffusion hypothesis; *Yamamura, 1999*). The type of surrounding crop has an impact on cabbageworm density (*Maguire, 1984*) and distance between cruciferous patches influences egg density of cabbage white butterfly (*Fahrig & Paloheimo, 1988*). Floral resource availability can also be important to the adult butterflies (*Curtis et al., 2015*).

Here, using the Asian honey bee (*Apis cerana*) and cabbage white butterfly (*Pieris rapae*), I investigated how environmental features within and surrounding urban farmlands in a megacity of China influenced the abundance of these two species, during winter in which Brassicaceae crops are of particular prevalent. *A. cerana* is small native species in southern and southeastern Asia. While the hive of *A. cerana* produces lower amount of honey than that of *A. mellifera*, *A. cerana* is more efficient in pollinating various fruits and vegetables (*Partap, 2011*). I focused on farmlands largely cultivating vegetables, which are dominant agricultural products of urban farming. I expected that the percentage of flowering plants and Brassicaceae crops would positively affect *A. cerana* and *P. rapae*, respectively. Given that these farmlands are located in urban areas mostly filled with buildings and houses, landscape weedy vegetation (spontaneous vegetation in vacant lots or parks) as well as available farmlands could also be critical to both species. Crop diversity could negatively affect *P. rapae* as found in the effect of crop diversification on insect pests in general (*Hooks & Johnson, 2003*; *Beillouin et al., 2021*). Alternatively, the effect may not be significant because urban farmlands in this study region often contain more diverse crops than in rural agricultural lands, and predator/parasitoid abundance is low in winter.

## MATERIALS & METHODS

### Study area

Guangzhou is the capital city of Guangdong Province, China (Fig. 1) and has a population around18 million. It also lies within the Indo-Burma Biodiversity hotspot region, the most urbanized area among biodiversity hotspot regions in China (*Güneralp & Seto, 2013*). Guangzhou has a subtropical monsoon climate: warm and dry winter and hot and humid summer, with a mean annual temperature of 22.2 °C (https://en.climate-data.org). A warm winter (mean temperature in January = 14.2 °C) allows for farmers to grow crops year-round. Although winter is relatively drier than other seasons, humidity is not low (60–71% between December and February). During the past two decades, farmlands between the inner city and the outskirts have drastically decreased due to urban expansion and economic development (*Shi & Shi, 2020*).

I selected 33 farmlands in 2020 and 57 farmlands (23 from previous year) in 2021 (Fig. 1). Several farmlands chosen in 2020 could not be surveyed in 2021 due to development,

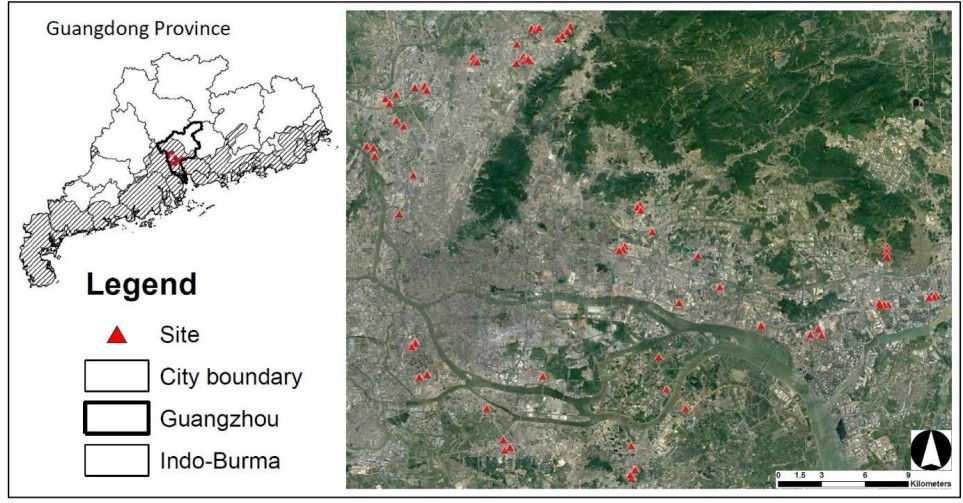

**Figure 1** **Study sites established in the city of Guangzhou, Guangdong Province, China.** All sites are located in the Indo-Burma Biodiversity Hotspot (Indo-Burma) areas in Guangzhou. Number of sites used for each taxon and year are variable. Satellite image source: Google Earth (https://earth.google.com/), Maxar Technologies 2022, CNES/Airbus 2022.

abandonment, and too-early growth stage of crops (very small seedlings). The size of all farmlands was ≥ 7,850 m$^2$. Within a farmland, I established a 50 m-radius area (site, hereafter) that was also used for bird surveys in another study (*Lee, Chen & Zou, 2022*). All samplings were performed within the 50 m-radius area during January each year. *P. rapae* was surveyed at 90 sites (32 in 2020 + 58 in 2021) across two winters: the distance between the closest two sites was 1.87 ± 1.09 km (mean ± standard deviation) in 2020 and 0.82 ± 1.18 km in 2021. Bee sampling was conducted at 58 sites (25 sites in 2020 and 33 sites in 2021).

I did not distinguish between urban and peri-urban agriculture, including farmlands located in the inner city as well as close to the city boundary (*Mougeot, 2000*). However, I emphasize that all sites were in urban areas; the proportion of built area (mostly buildings and houses) within a 500 m-radius area of sample sites averaged 61 ± 13.8%, consistent with the criterion of urban areas containing >50% built area (*Marzluff, 2001*).

## Cabbage white butterfly *Pieris rapae*)

*P. rapae* was surveyed twice (≥ 12 days apart) using a transect method: one survey in the morning (between 9–11:30 am) and one in the afternoon (between 1:00-4:00 pm). During each survey, I randomly placed two 50 m-line transects along a walk path within a site. However, if Brassicaceae and non-Brassicaceae crop areas were clearly divided, I placed one transect close to more Brassicaceae crops and the other near non-Brassicaceae crops. I also changed the locations of transects between surveys. I walked the transect at a rate of 5 m per minute, and counted *P. rapae* individuals detected within a 10 m × 5 m × 5 m imaginary window (5 m to each side of the transect, 5 m from the ground, and 5 m in front). Although a 5 m width is commonly used, I expanded the width of the imaginary

window because *P. rapae* was often active slightly farther from the edge of path and moved along the row crops, which were not easy to access because of irrigated water and farmer's attendance at some sites. Special care was taken not to count the same individuals twice while walking along the transect. During the first winter, I also carried out another survey using an area search method to verify the transect method. I quickly walked around 50% area of a site, *e.g.*, north-east-south side for 1 min, and counted *P. rapae*. This scanning process was repeated three times at different sides of the site, with 3 min between surveys. Data collected from transect method and area search methods were highly correlated: Pearson's correlation (r) between maximum counts of each data was 0.94 ($P < 0.001$), indicating that the transect survey data adequately represented *P. rapae* abundance in this study.

### Bee sampling

To capture bees, I used three-colored (blue, white, and yellow) pan traps (*Campbell & Hanula, 2007*). Three sample stations were established >30m apart from each other within a site. Sample stations were placed randomly from the center of site. However, most locations were restricted by presence/absence of flowering plants (crops and weeds), farmer's permission, and accessibility (avoiding deep furrows filled with water). I installed a set of three-colored pan traps at each sample station. I attached pan traps to a pole ∼0.8 m high, which was tall enough to allow bees to see them, and placed them near flowering crops or weeds if they were present. Pan traps were $1/2 - 2/3$ filled with soapy water (10 drops of dish soap per liter) to remove the surface tension. All traps were retrieved 2 days after installation. While I brought all bees captured to a lab, washed, and preserved in 70% ethyl alcohol for identification, I counted only *A. cerana* because they were the main bee species captured and abundant enough to conduct analysis.

### Environmental data at fine and local scales

Local scale environmental data were collected within a site, that is, "site" represents the local scale of this study. At each site, crop and non-crop features such as storage house, path, herbaceous weeds, trees/shrubs, and open water (small pond and water channel) were identified and marked on a printed satellite image downloaded from Google Earth (https://www.google.com/earth/). Many farmlands were composed of highly diverse crop species, but each crop was cultivated in similar-sized rows. Thus, I divided the site into several blocks. I identified crops and counted the number of rows of each crop within a block. I summed all counts by block. In a block, I calculated the relative frequency of each crop from the sum and converted that to "area" by multiplying it by the size of the block. The sum of the crop's area across all blocks represented total area of the crop within the site. All non-crop features and blocks were delineated in ArcGIS using georeferenced Google Earth images as a base map.

I calculated the percentage of weedy vegetation (weed50) and cruciferous crops (Brassicaceae; the mustard family). Crop diversity (cropdiv50) was calculated as a Shannon-Wiener diversity index. While over 120 crops were found across sites, several crops were very minor or variants of the same crop, sharing similar biological and ecological characteristics.

I also found cases of misidentifications when crops look similar, especially when they belong to the same genus or family. To minimize any bias associated with misidentification and very minor crops, I used the family of crop for the calculation of cropdiv50 (Table S1 for the list of crop family). At the family level, all sites had $\geq 2$ crops and approximately 95% of sites contained at least six crops, averaging $13.9 \pm 5.4$, and ranging from 2 to 24.

For *A. cerana*, crop surveys were also performed at a fine scale, a 5 m-radius area surrounding a sample station. Within the 5 m-radius area, crops were identified and the percentage of each crop was visually estimated. Similar to cropdiv50, I calculated fine-scale crop diversity based on crop species (cropdiv5). In 2021, the percentage of flowering plants (flower) including crops and weeds was also estimated at the fine scale.

### Landscape data

I chose two landscape-scale sizes by considering flight distances of *P. rapae* and *A. cerana* (*Jones et al., 1980*; *Dyer & Seeley, 1991*), the logistics of creating a land cover map, and the matrix context of study sites, selecting 500 m-radius and 1 km-radius areas surrounding the center of site. The 500 m-radius area was used to examine compositional aspect of the landscape matrix in which a farmland was embedded, and the 1 km-radius area was used to characterize the spatial configuration and fragmentation of farmlands within a landscape.

To generate the land cover map, I downloaded satellite images from Google Earth and georeferenced them. All images were taken between August, 2019 and February, 2021. Using the georeferenced images as a base map, I delineated three land cover types within the 500 m-radius area in ArcGIS: vegetable-dominant farmland (crop cover), weedy vegetation (mostly spontaneous herbaceous vegetation in vacant lots and construction sites), and built structure (building, house, road, and any impervious surface). I then calculated the percentage of farmland or crop cover (crop500) and weedy vegetation (weed500). I also delineated farmlands within the 1 km-radius area and calculated patch density (pd1000) and edge density of farmlands (edge1000) using Fragstats v 4.2.1 (*McGarigal, Cushman & Ene, 2012*). Although these indices may be the simplest measure of the spatial configuration of habitat patch (*McGarigal & Marks, 1995*), they can affect diversity and abundance of pollinators and butterflies by increasing edge habitats or facilitating movements between patches (*Flick, Feagan & Fahrig, 2012*; *Hass et al., 2018*; *Martin et al., 2019*; among others). Edge density is the same as the total edge length in this study because landscape sizes were identical across all sites.

### Statistical analysis

I pooled 2 years of data together for *P. rapae*; even if the same site was surveyed both years, each year's data were considered independent because local scale features such as crop diversity and amount of weedy vegetation differed between two winters at the same site. Any potential bias associated with this approach, *i.e.*, year effect was also examined before final analysis (see below). Among environmental variables, edge1000 was highly correlated with pd1000 and crop500 ($r = 0.83$ and $0.68$, respectively, $P < 0.001$ in *P. rapae* data), which increased the variance inflation factor of these variables. Thus, edge1000 was not included in analyses.

I selected the maximum count of *P. rapae* between two transects and summed it over two visits at each site. The sum was used as the "abundance" of *P. rapae*. I log(x+1)-transformed the abundance of *P. rapae* to minimize potential bias caused by high abundance values and overdispersion that can affect type I error. I first tested whether survey year (2020 or 2021) had an effect on *P. rapae* abundance. I compared two generalized linear models (GLMs) with Gamma distribution, *i.e.,* intercept-only model *vs* model with a year variable using the likelihood-ratio test. It showed no significant difference between two models ($\chi^2$ = 0.006, $P = 0.937$), suggesting that *P. rapae* abundance was not affected by year-related variations. Thus, I constructed four GLMs without "year" variable ( Table S2): null model (intercept-only model), local model (three local variables—Brassicaceae, cropdiv50, weed50), landscape model (three landscape variables—crop500, weed500, pd1000), and full model (three local and three landscape variables). These four models were compared based on their AICc (Alkaike Information Criterion adjusted for small sample size) values following an information-theoretic model-selection approach (*Burnham & Anderson, 2002*). In the approach, a model with a lowest AICc is ranked as a top model and represents the best-supported model. However, other models with Δ AICc (AICc difference from top model) <4 are also considered plausible models to explain variations in the data. Thus, I used a top model to make inferences if there were no competing models. Otherwise, I performed model averaging on plausible models (Δ AICc <4) and used model-averaged parameter estimates for inferences ("MuMIn" package; *Bartoń, 2022*).

I summed *A. cerana* individuals caught in a set of pant traps at each sample station, which was considered as a unit for the analysis. The sum represented the abundance of *A. cerana* per sample station. Environmental data for *A. cerana* included fine-scale data. However, flower covers at the fine scale was collected in 2021 only and thus I performed two separate analyses: the one with the data of two years (2020 and 2021) and the other with 2021 data only. I built a total of eight GLMs with negative binomial distribution for each analysis using a combination of scales (Table S3). The survey year had a significant effect on *A. cerana* abundance and thus year was incorporated into all models that used both 2020 and 2021 data. The abundance data of *A. cerana* had zero values; while zero-inflated Poisson and negative binomial distributions can be used in this case, I chose negative binomial distribution because of overdispersion in the GLMs with zero-inflated Poisson. Model selection and averaging were carried out as for *P. rapae*.

Spatial dependence of bee and butterfly counts was also examined by conducting Moran's I test on the residuals of the full models of *P. rapae* and *A. cerana* ("ape" package; *Paradis & Schliep, 2019*): *P* >0.1 in all cases, indicating that spatial dependence was negligible. I checked other regression model assumptions such as homoscedasticity and overdispersion ("DHARMa" package; *Hartig, 2022*) and did not find any cases of significant violation. The values of variance inflation factor were <2.5, suggesting little issue of multicollinearity.

## RESULTS

A total of 1,189 *P. rapae* was observed across all sites. Mean abundance was 13.21 butterflies per site with a SD of 10.16, ranging from 1 to 39. The abundance of *P. rapae* was strongly

**Table 1 Model selection results of Pieris rapae based on Akaike information criterion value adjusted for small sample size (AICc).** Note a big difference in AICc values (ΔAICc) between the top model and other models. Null model is an intercept-only model, whereas full model includes both local- and landscape-scale variables.

| Model | DF | LogLik | AICc | Δ AICc | AICc weight |
|-------|-----|---------|-------|---------|-------------|
| Full | 8 | −81.506 | 180.8 | 0.00 | 0.988 |
| Landscape | 5 | −89.493 | 189.7 | 8.91 | 0.011 |
| Local | 5 | −95.622 | 202.0 | 21.17 | 0.000 |
| Null | 2 | −113.276 | 230.7 | 49.90 | 0.000 |

Notes.
  Abbreviation: DF, degree of freedom; LogLik, log-likelihood.

associated with a combination of local and landscape variables: the full model showed the lowest AICc, and Δ AICc of other models was >9 (Table 1).

Local crop diversity had a positive effect on the abundance of *P. rapae*, whereas landscape-level farmland and weedy vegetation covers affected *P. rapae* abundance negatively (Fig. 2 and Table S4). The fragmentation of farmlands, *i.e.,* patch density, was not associated with *P. rapae* abundance given wide 95% confidence intervals across 0 and very low estimate (Table S4). Although percent cover of cruciferous crops tended to have a positive impact on *P. rapae* abundance, the effect was not as strong as other significant variables (Table S4).

For two winters, 265 *A. cerana* individuals were captured. While more individuals were caught in 2021 than 2020, *A. cerana* abundance varied by sample stations: 1.15 ± 2.11 (mean abundance ± SD), ranging from 0 to 14 in 2020, and 1.82 ± 2.03, ranging from 0 to 11 in 2021. Of 8 GLM models, 4 models were selected as plausible models in 2021 data analysis and 2 models in combined years data analysis (Table 2).

Effects of weedy vegetation on *A. cerana* abundance were consistent across two winters given that the 95% confidence intervals did not include 0 or slightly overlapped with 0. The effects also depended on scale: the abundance of *A. cerana* was affected by weedy vegetation cover negatively at the landscape scale but positively at the local scale (Fig. 3 and Table S4). In combined years data analysis, increasing percent cover of farmland at the landscape scale was positively associated with increasing *A. cerana* abundance. *A. cerana* abundance was also low in a landscape with more fragmented farmlands. The percentage of flowering plants had a strong positive effect on *A. cerana* abundance in 2021: its parameter estimate (*i.e.,* effect) was ≥ 1.5 higher than the estimates of other variables associated with *A. cerana*, suggesting the amount of floral resource at the fine scale could be critical to *A. cerana* (Fig. 3).

## DISCUSSION

The results of this study reveal variation in environmental factors at relevant spatial scales associated with a host-specific pest, *i.e.*, *Pieris rapae*, and a pollinator, *i.e.*, *Apis cerana*. *P. rapae* was more related to crop cover at the landscape scale (negatively), whereas *A. cerana* was strongly related to flower cover at the fine scale and weedy vegetation at the local

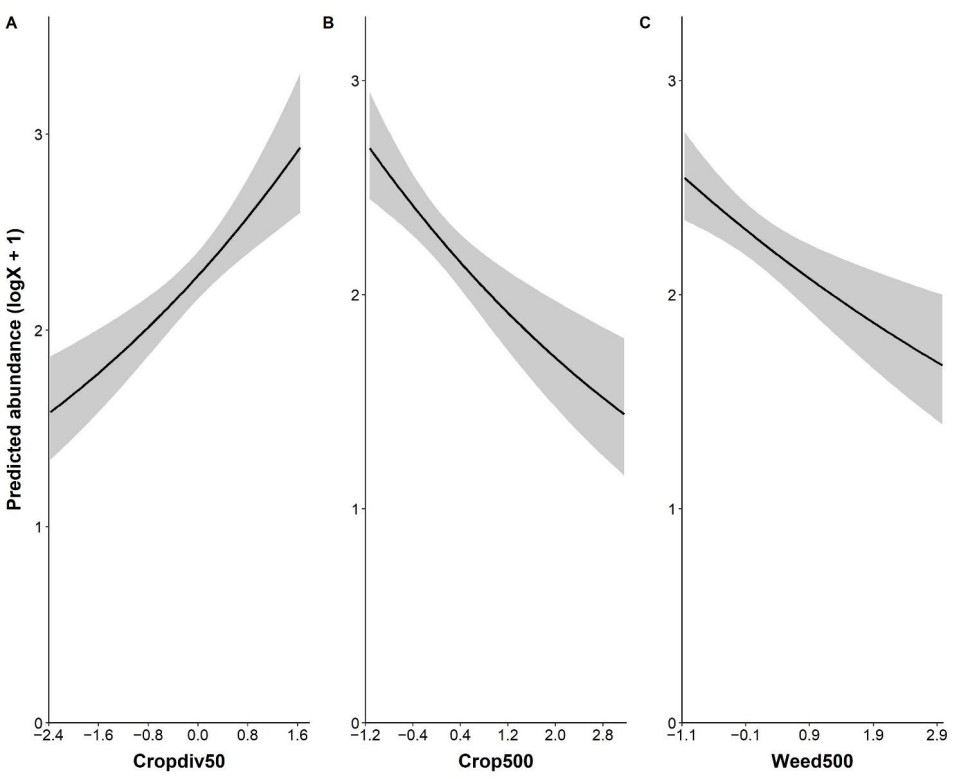

**Figure 2  Significant relationship between the abundance of *Pieris rapae* and environmental variables: local crop diversity (cropdiv50, A), landscape crop cover (crop500, B), and landscape weedy vegetation cover (weed500, C).** $Y$-axis represents predicted abundance that is $\log(x+1)$ transformed. $X$-axis shows standardized values of each environmental variable.

scale (positively). Both species also responded negatively to weedy vegetation cover at the landscape scale. These patterns can provide insight into how small urban farmlands can be managed to improve ecosystem service, *i.e.,* pollination by *A. cerana*, while minimizing ecosystem disservice, *i.e.,* crop damage by *P. rapae*.

The resource concentration hypothesis predicts high abundance of host-specific herbivorous insects at their host plant rich patches, *i.e.,* dense, large, or pure stands of host plants (*Root, 1973*). I expected a positive relationship between *P. rapae* abundance and percent cover of Brassicaceae, and I found a tendency for this relationship. However, landscape-scale crop cover was the environmental variable most strongly associated with the abundance of *P. rapae*. Certainly the presence or absence of Brassicaceae in surrounding farmlands could influence *P. rapae* distribution. If Brassicaceae cover is low in a landscape with otherwise high crop cover, we might expect a negative association. Although I lack data on Brassicaceae cover at the landscape scale, local Brassicaceae cover was not correlated with landscape crop cover. Also, almost all farmlands surveyed contained Brassicaceae: of 90 sites, 44 included >20% Brassicaceae at the local scale and only seven had <5% Brassicaceae. Thus, the negative response of *P. rapae* is unlikely related to the amount of landscape Brassicaceae cover. *Matteson & Langellotto (2012)* found that *P. rapae* spends

**Table 2 Summary of model comparisons of *Apis cerana*.** Null model is an intercept-only model, whereas full model includes variables of all three scales. A combination of two spatial scales is denoted with "+": for example, Fine+Local model contains variables at fine and local scales. Note that fine-scale flower cover was considered in 2021 only. Models in "Both" year are constructed using two years of data, *i.e.*, 2020 and 2021.

| Year | Model | DF | LogLik | AICc | Δ AICc | AICc weight |
|------|-------|-----|--------|------|--------|-------------|
| 2021 | Full | 9 | −163.529 | 347.1 | 0.00 | 0.444 |
| | Fine+Local | 6 | −167.580 | 348.1 | 0.99 | 0.270 |
| | Fine+Local | 4 | −170.216 | 348.9 | 1.78 | 0.182 |
| | Fine+Landscape | 7 | −167.387 | 350.0 | 2.92 | 0.103 |
| | Local+Landscape | 7 | −172.985 | 361.2 | 14.12 | 0.000 |
| | Local | 4 | −176.948 | 362.3 | 15.24 | 0.000 |
| | Landscape | 5 | −176.131 | 362.9 | 15.83 | 0.000 |
| | Null | 2 | −179.720 | 363.6 | 16.49 | 0.000 |
| Both | Local+Landscape | 8 | −280.330 | 577.5 | 0.00 | 0.582 |
| | Full | 9 | −279.815 | 578.7 | 1.20 | 0.320 |
| | Landscape | 6 | −284.734 | 582.0 | 4.44 | 0.063 |
| | Fine+Landscape | 7 | −284.620 | 583.9 | 6.38 | 0.024 |
| | Local | 5 | −288.486 | 587.3 | 9.79 | 0.004 |
| | Null | 3 | −291.062 | 588.3 | 10.73 | 0.003 |
| | Fine+Local | 6 | −288.123 | 588.8 | 11.21 | 0.002 |
| | Fine | 4 | −291.054 | 590.3 | 12.81 | 0.001 |

**Notes.**
Abbreviation: DF, degree of freedom; LogLik, log-likelihood; Δ AICc, AICc difference from top model.

less time in urban gardens in a landscape with more green spaces, which could lead to low detection of cabbage butterfly in these gardens. I often observed more *P. rapae* individuals and their mating, landing, and oviposition behaviors at farmlands isolated in a landscape dominated by built structure. Considering that crop and weedy vegetation covers are part of green spaces, the responses of *P. rapae* to both covers parallels the previous finding. Increasing crop covers in a landscape likely diffuses *P. rapae*, lowering abundance per farmland. It may also interrupt visual cues and consequently have a negative impact on their ability to search host and floral resources because *P. rapae* depends on vision to locate these resources (*Hern, Edwards-Jones & McKinlay, 1996*).

The positive effect of crop diversity is somewhat unexpected. Polyculture stands often show lower density or higher mortality of pest insects compared to monoculture stands (*Altieri et al., 1978*; *Russell, 1989*; *Letourneau et al., 2011*; *Iverson et al., 2014*; among others). The diversification of cruciferous crops through mixing or intercropping can effectively control pest insects due to higher abundance and richness of parasitoids/predators, visual camouflage effect, and masking effect of host plant odors, among other mechanisms (*Finch & Collier, 2000*; *Hooks & Johnson, 2003*). The positive effect I observed may be affected by the season of survey. Although winter is relatively warm and dry in southern China, the abundance and richness of predator and parasitoid insects are lower than other seasons such as Spring (M-B Lee, 2020-2021, pers. obs.). *Lowenstein & Minor (2018)* reported different trends in abundance between herbivores including *P. rapae* and predator/parasitoid

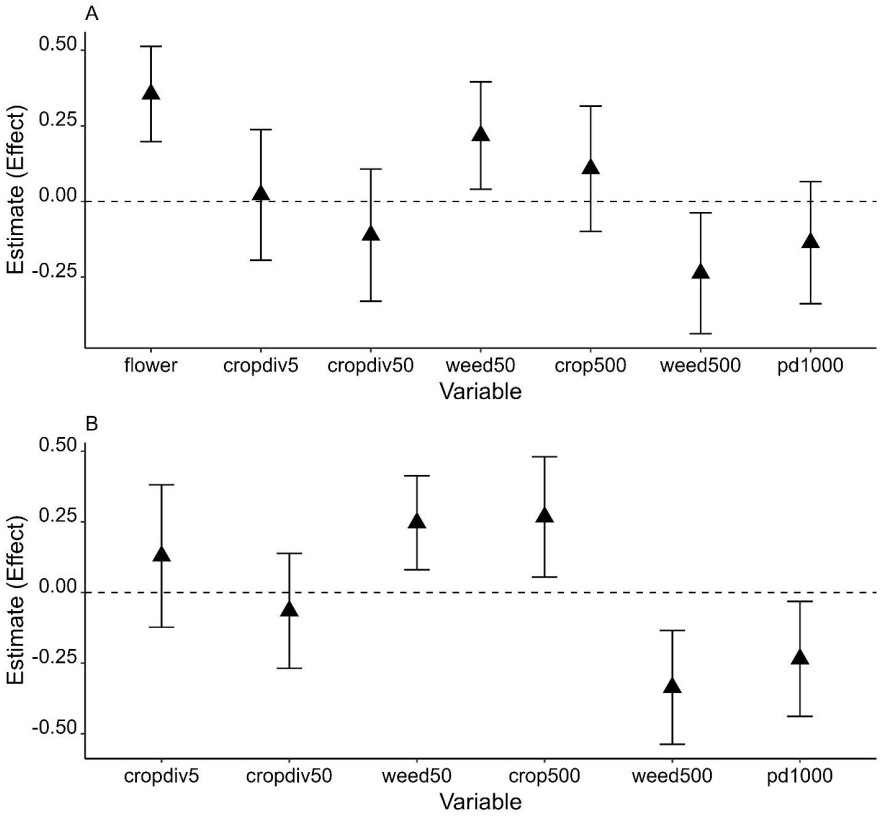

**Figure 3** **Model averaged parameter estimates and their 95% confidence intervals (error bars on the graph) from the analysis of *Apis cerana* using 2021 data (A) and two years of data (B).** *X*-axis shows all environmental variables at each of three spatial scales: fine scale variables, flower (flower cover) and cropdiv5 (crop diversity); local scale variables, cropdiv50 (crop diversity) and weed50 (weedy vegetation cover); landscape scale variables, crop500 (crop cover), weed500 (weedy vegetation cover), and pd1000 (patch density).

insects at urban gardens and farms in Chicago, IL, USA; areas with high predators and parasitoids tended to have low herbivores. Compared to large-scale rural agricultural lands, urban farmlands have also more diverse crops despite their small size. Brassicaceae crops are cultivated at most of these farmlands in the study area. This creates a spatially and temporally heterogeneous environment due to variations in harvest time and growth stage between crops, even between Brassicaceae crops. These conditions may benefit *P. rapae* by providing host and floral resources, shelters, or mating chances for a longer period during the winter season. I also noticed seasonal variations in crop composition, although crop diversity at each study site did not change significantly between seasons. For example, in spring, the proportion of Brassicaceae crops declines but other crops (especially vine crops, *e.g.*, bean and gourd family crops) increase and flowering weeds are abundant. *P. rapae* may respond differently to crop diversity in spring, but the extent of possible seasonal variations in the responses of *P. rapae* requires further research.

Bee populations are strongly regulated by food resource availability (*Roulston & Goodell, 2011*). The establishment of semi-natural vegetation, especially planting flowering herbaceous plants at field edges (*i.e.,* flower strips) is often recommended to promote crop pollination and yield by insect pollinators (*Blaauw & Isaac, 2014*; *Williams et al., 2015*; *Shutter, Albrecht & Jeanneret, 2018*; but see *Nicholson et al., 2019*). One recent meta-analysis shows that crop pollination decreases with increasing distance to floral resources, indicating the importance of floral resources to pollinators (*Albrecht et al., 2020*). Crop diversity and semi-natural vegetation at landscape scales also increase bumble bee density in wheat-dominant agricultural areas (*Raderschall et al., 2021*). The result of the current study, *i.e.,* the strong effect of percent cover of flowering plants on bee abundance, partly supports the general trend. While I do not have data on the diversity and abundance of flowering plants at local and landscape scales, the result highlights the benefit of even small amount of flowering crop and weed to honey bees in urban environments, especially during the winter in which natural floral resources may be scarce. This could also explain the different effects of weedy vegetation cover at local and landscape scales. The subtropical climate in southern China enables framers to cultivate vegetable crops year-round. While some weedy vegetation showed withering in winter, the condition was less severe at urban farmlands than some vacant lots (*e.g.*, construction sites and abandoned lands), which is likely affected by differences in water availability. There were also more weed plants blooming at urban farmlands based on my observation.

One potential concern is the quality of floral resources in these urban farmlands because most common weed plants flowering in the study area are non-native plants, particularly Bidens species. There are few studies comparing nutritional values of nectars to bees between crops and weed plants in this region. However, honey bees are generalists, visiting a wide range of flowering plants, and require different diets to maintain colony health (*Requier et al., 2015*). During times of food shortage, weed plants and non-native plants can be an important component of the diets of *A. mellifera* and *A. cerana,* respectively (*Requier et al., 2015*; *Koyama et al., 2018*). *A. mellifera* also forages on flowers of Bidens species (*Kajobe , 2007*). Floral resource use and preference by bees are significantly associated with dominant plants and affected by season as floral resource availability changes throughout the year (*Lowe et al., 2021*). Most flowering plants at sample sites in winter are weed plants and several crops that belong to the genera Brassica, Chrysanthemum, and Allium. Weed plants can be one of major floral resources to *A. cerana* in winter regardless of their nutritional values. However, with seasonal variations in the diversity and abundance of floral resources, *A. cerana* may prefer certain crops or weed plants in other seasons such as spring, when trees and shrubs as well as more crops bloom. The degree of effects of flower strips on local bee abundance and diversity can also depend on the interaction between characteristics of flower strips and floral resources available in surrounding landscape (*Scheper et al., 2015*). Similarly, it is possible that the strong response of *A. cerana* to fine-scale flower cover could be influenced by the total amount of floral resources within farmland and landscape, which will vary by season.

It is noteworthy that patchy density of farmland had a negative impact on *A. cerana* abundance weakly or significantly. This suggests that *A. cerana* population may be

susceptible to farmland fragmentation to some degree. While the honey bees' response to habitat fragmentation is rarely explored, one recent study shows that density of green patch in urban areas can have an indirect negative effect on total abundance of bees by negatively influencing flowering plant richness (*Theodorou et al., 2020*). In the current study, farmland fragmentation, *i.e.,* patch density of farmland is highly positively correlated with total edge length of farmlands. Banana plants are common along the boundaries of some farmlands and used as a fence. Banana flowers may be an important nectar and pollen source for some managed honey bee (*e.g.*, hybrid Carniolan honey bee, *A. mellifera carnica* Pollmann) when they bloom (*Taha, Taha & AL-Kahtani, 2019*). However, winter is not a bloom season of banana plants in the Guangdong province. There are also a variety of flowering trees, weeds, and crops that can be used by bees in other seasons. Thus, banana-dominant vegetation at the edge of farmland likely reduce the space that could be covered with weedy plants and consequently floral resources available to *A. cerana*, especially in winter. Fragmented farmlands may be also managed intensively: for example, frequent weed removal at the farmlands can decrease local weedy vegetation cover. However, these explanations remain speculative until we have multiple studies performed in other regions and sufficient data to compare them for better understanding of the relationship between *A. cerana* and farmland fragmentation.

## CONCLUSIONS

Overall, the findings of current study can provide a scientific basis for urban planners, policy makers, and farmers to consider in the management of urban farmlands across scales to enhance the sustainability of these farmlands, particularly in large subtropical city like Guangzhou. At the farmland (local) scale, cultivating too diverse crops may not be recommended as it can increase the risk of crop damage by *P. rapae*. However, the risk can be alleviated at farmlands embedded in a landscape with more overall crop cover, which may partly benefit *A. cerana* as well. Retaining non-crop area such as weedy vegetation patches or small portion of flowering plants including crops within a farmland can be an effective practice to maintain *A. cerana* population and promote pollination. It also benefits birds as local weedy vegetation has a positive effect on winter bird diversity at urban farmlands (*Lee, Chen & Zou, 2022*). With growing urban expansion, farmlands in China have been significantly converted into built area, leading to a decline in net primary productivity of cropland (*He et al., 2017*) despite farmland protection policy restrictions on development. Farmlands in a city are not exceptional. In major cities in China, new development largely spreads from the edge of city or occurs at old villages within a city (*Tu et al., 2021*), lowering crop cover in surrounding landscapes and isolating farmlands (*Liang et al., 2015*). Policy makers and urban planners need to consider preserving farmlands, especially at the city fringe or suburb for permanent agricultural activities. Priority may be given to farmlands embedded in landscapes with relatively high crop covers. It is also critical that city government monitors potential environmental problems associated with urban farmlands. For example, most of these farmlands, especially ones next to main districts where development and population are concentrated, use waste water. The average

contents of heavy metals in agricultural soil in Guangzhou and Foshan cities are higher than background values of Guangdong province (*Xiao et al., 2020*). While the levels of metals are still lower than national standard values, the ecological risk of heavy metal pollution in these cities is not negligible. Combined with further research on seasonal variations in the patterns found in current study, the assessment of environmental quality of urban farmlands can inform efforts to improve the sustainability of urban agroecosystem. At the same time, it is important to note that the findings of current study are based on a single study conducted in one megacity. Although the findings offer valuable insights on environmental factors associated with insect pollinators and pests at urban farmlands, more studies are needed in other tropical/subtropical cities to draw general conclusions.

# ACKNOWLEDGEMENTS

I would like to thank Y. Zhang and D. Cheng for their assistance in collecting local environmental data as well as honey bee data, D. Chen for handling logistic issues, and J. Rotenberry for edits and comments on the draft of this manuscript.

## Funding
This study was supported by the Guangdong Academy of Sciences Special Project of Science and Technology Development 2020GDASYL-20200103089. There was no additional external funding received for this study. The funders had no role in study design, data collection and analysis, decision to publish, or preparation of the manuscript.

## Grant Disclosures
The following grant information was disclosed by the author:
Guangdong Academy of Sciences Special Project of Science and Technology Development: 2020GDASYL-20200103089.

## Competing Interests
The authors declare there are no competing interests.

## Author Contributions
- Myung-Bok Lee conceived and designed the experiments, performed the experiments, analyzed the data, prepared figures and/or tables, authored or reviewed drafts of the article, and approved the final draft.

## Data Availability
The raw data are available in the Supplemental Files.

## Supplemental Information
Supplemental information for this article can be found online at http://dx.doi.org/10.7717/peerj.15725#supplemental-information.

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
