# Peer review of "Environmental factors affecting honey bees (Apis cerana) and cabbage white butterflies (Pieris rapae) at urban farmlands"

_PeerJ, doi:10.7717/peerj.15725_

## Round 0.1 · original submission · Minor Revisions

The originality and the importance of this study are underlined by all the referees, but there are some minor and customary issues that you have to revise before the final acceptance. I am sure that the referees' suggestions will improve the general quality of your manuscript.

·

Basic reporting

The manuscript is generally well written, clear and concise with sufficient background and literature references. Raw data were shared and figures and tables are clear. I did have a couple of questions/suggestions:

I was unfamiliar with the usage of "built-up" areas throughout and wondered if a more common usage might be "more highly developed" areas. Lines: 43, 131-2, 171, 207, 312, 393.

Line 197 I think you meant chose not choose

Lines 406-409 suggest changing "can contribute to improve" to "can inform efforts to improve"

Experimental design

I found the investigation to be focused and carefully conducted. The analyses seemed appropriate. I did have a question about how the area of landscape level weedy vegetation was ascertained. Perhaps a bit more clarity of what constituted weedy vegetation and its status during the sampling periods would be helpful given the somewhat surprising (to me) negative association with bees.

Validity of the findings

Conclusions are consistent with results presented.

Additional comments

This manuscript reports the results of a carefully conducted and thoroughly analyzed study with findings that can inform urban planning. I recommend publication with minor edits.

Reviewer 2 ·

Basic reporting

I thank the author for the interesting research article!

The author gives a good introduction to the subject and elaborates on the importance of the research using relevant and current literature. The manuscript is well structured and includes figures and tables to visualize relevant information. However, the map included in figure 1 could have a higher resolution to see the details/sites more clearly. For a better overview, it would be helpful to list the environmental variables in a table, sorted by the scales. Raw data is provided.

The language is professional and unambigous. However, in some passages the English language/ expression in general could be improved to ensure comprehensibility. Regarding the language following lines have come to my attention:
line 21: "... city citizens..."
line 101: ".. farmlands are in urban areas..." ( I would suggest "located in urban areas")
line 112: "peoples"
line 116: "... located in city center..."
line 161: "...a pole at 0.8 m high..." ( I would suggest: "to a pole ~0.8 m high")
line 170.
Futher, I noticed spelling mistakes in the scientific names of two bee species in line 367.
I would prefer/suggest using the scientific names Pieris rapae and Apis cerana throughout the article, rather than the colloquial names.

Experimental design

The research question is clearly defined and the relevance is highlighted in the introduction, which also provides background information on the timeliness of the research topic. Nevertheless, I would recommend further emphasizing how this particular study can help fill an existing research gap and provide important insights for management practices in urban agriculture.

The reserach has been conducted to a high technical and ethical standard. All methods are described in detail, allowing for replication of the study. Sampling methods and statistical analyses are appropriate and well described. I appreciate the extensive sampling effort (high number of sites) and the inclusion of a range of environmental parameters, even tailored to the ecological characteristics of the species studied (activity radius). For clarification, I would recommend to add a short and clear definition of "built-up cover" (line 130). As stated in 1., I would suggest listing all environmental factors in a table.

The satellite images for environmental data were taken between 2018 and 2020. Are the images still suitable to depict the state of the surrounding in the years the survey was conducted (2020 and 2021)? A brief comment on this would be appreciated.

Validity of the findings

With the research presented in the manuscript, the author has contributed to fill a current gap in understanding the effects of environmental factors on an important pollinator species (Apis cerana) and a common pest (Pieris rapae) in farmlands within an urban context. Against the backdrop of increasing urbanization, this study can provide an important basis for future research in urban agriculture.

The underlying data has been provided and is statistically sound.

The conclusion is well connected to the original research question and takes up the initial hypotheses of the author. The author mentions how findings can help conceptualize suitable management practices for sustainable urban agriculture. However, the conclusions should explicitly note that the results of this single study should not be overstated and replication is needed before general conclusions can be drawn (line 385).

Additional comments

Although the author mentions the effects of patch density on CWB (no effect) and honey bees (slight effect) in the abstract, this result is not mentioned in the results section nor taken up in the following discussion. In my opinion it would make sense to address this result (as it is presented in the abstract). Same applies to the predictor edge density, which was used as an environmental variable, but never mentioned in the results or discussion.

For the introduction, I would suggest adding some information(one sentence)/refences on the importance of honey bees in particular as this research focuses on one domesticated species and does not address wild bees or bee diversity in general.

I find the explanation for the negative response of Pieris rapae to the cover of weed and farmland very intriguing (line 308 ff)! At sites with less suitable green space in the vicinity, CWB may occur in greater numbers or remain for a longer period of time, as these areas may serve as refuges with valuable resources.

Reviewer 3 ·

Basic reporting

I would like to thank you for the opportunity to prepare the review for the paper titled Environmental factors affecting insect pollinators 1 and Pests at urban farmlands: a case study with honey bees and cabbage white butterflies.

The manuscript is well done and complete although I strongly suggest the author change the writing style. It would be desirable that the author did not write in the first person but being a scientific article, he wrote in an impersonal way, possibly using the passive voice.
The key concept is well formulated. The research topic in that formulation is very interesting and not much developed in other previous reviews or articles.

Experimental design

Unfortunately, I need to present critical remarks. I don't deny the validity of the concept and the quality of the paper but some methods would better explain. In particular, the use of Shannon index at the family level to calculate cropdiv50. Also, the significance of The AICc (Alkaike Information) would be clarified better.
I also suggest trying to add some more recent literature to compare the data. In particular the number of insect captures seems very low could you provide also climate data not only as average temperature but also the minimum and maximum as well relative humidity. Could you compare the insect capture in your urban areas with other findings?
In addition, the authors should report also other data other if possible economic data.

Some other small suggestions are listed below.
- Use the acronyms of cabbage white butterflies (CWB) the first time that you refer to them.
- Line 78 refers here which bees you consider.
- Line 98 Explain in the introduction a little bit more about the condition of winter in the studied areas.
- 125-127 please rephrase the concept is not really clear.
- 152-153 Delete the sentence that starts with "I....." It is a consideration, not a material and method.
- Please put the word Apis in italics
211-214 this sentence starting with “Although these …” is not appropriate here please move in the introduction or in the discussions.
265-266 please clarify the sentence is not clear.
283-285 Please explain better the meaning of the coefficients.

Validity of the findings

The findings are very interesting and well discussed.

---

## Round 0.2 · accepted · Accept

Both referees gave their final decision and agreed that the manuscript is greatly improved and it can be accepted in the present form

Reviewer 2 ·

Basic reporting

The manuscript is well written and structured with current literature references, sufficient background information and includes figures and tables to visualize relevant information. Raw data is provided.

Experimental design

All methods are described in detail, allowing for replication of the study. Sampling methods and statistical analyses are appropriate and well described.

Validity of the findings

No comment

Additional comments

The author has answered open questions and addressed comments in the manuscript, incorporating addtitional explanations.

Reviewer 3 ·

Basic reporting

The author improved the manuscript according to the suggestions of the reviewers.
The manuscript is well done and complete The key concept is well formulated. The research topic in that formulation is very interesting and has not developed much in other previous reviews or articles.

Experimental design

The author was rigorous enough to cover all the adequate breadth & depth. The methodology is adequately described and appropriate.

Validity of the findings

The results are well described and discussed. The findings are interesting and they go behind the state-of-art.